# Extraction of Bioactive Compounds for Antioxidant, Antimicrobial, and Antidiabetic Applications

**DOI:** 10.3390/molecules27185935

**Published:** 2022-09-13

**Authors:** Fatimah Saeed Aldughaylibi, Muhammad Asam Raza, Sumaira Naeem, Humera Rafi, Mir Waqas Alam, Basma Souayeh, Mohd Farhan, Muhammad Aamir, Noushi Zaidi, Tanveer Ahmad Mir

**Affiliations:** 1Department of Physics, College of Science, King Faisal University, Al-Ahsa 31982, Saudi Arabia; 2Department of Chemistry, University of Gujrat, Gujrat 50700, Pakistan; 3Department of Basic Sciences, Preparatory Year Deanship, King Faisal University, Al-Ahsa 31982, Saudi Arabia; 4Laboratory of Tissue/Organ Bioengineering and BioMEMS, Organ Transplant Centre of Excellence, Transplantation Research & Innovation (Dpt)-R, King Faisal Specialist Hospital and Research Centre, Riyadh 11211, Saudi Arabia

**Keywords:** antimicrobial, antidiabetic, antioxidant, medicinal plants, GC-MS

## Abstract

This study was designed to check the potential of secondary metabolites of the selected plants; *Citrullus colocynthis*, *Solanum nigrum*, *Solanum surattense,* *Calotropis procera*, *Agave americana*, and *Anagallis arvensis* for antioxidant, antibacterial, antifungal, and antidiabetic agents. Plant material was soaked in ethanol/methanol to get the crude extract, which was further partitioned via solvent extraction technique. GCMS and FTIR analytical techniques were applied to check the compounds responsible for causing antioxidant, antimicrobial, and antidiabetic activities. It was concluded that about 80% of studied extracts/fractions were active against α-amylase, ranging from 43 to 96%. The highest activity (96.63%) was exhibited by butanol fractions of *A. arvensis* while the least response (43.65%) was shown by the aqueous fraction of *C. colocynthis* and the methanol fraction of fruit of *S. surattense.* The highest antioxidant activity was shown by the ethyl acetate fraction of *Anagallis arvensis* (78.1%), while aqueous as well as n-hexane fractions are the least active throughout the assay. Results showed that all tested plants can be an excellent source of natural products with potential antimicrobial, antioxidant, and antidiabetic potential. The biological response of these species is depicted as a good therapeutic agent, and, in the future, it can be encapsulated for drug discovery.

## 1. Introduction

The utility and discovery of plants having bioactive ingredients are as ancient as their uses as medicines, food, and nutraceuticals [1]. A huge diversity of plants is being used by native people for the curing of many diseases throughout the whole world. It is documented that about eighty percent of the community of developing countries is reliant on medicinal plants for health maintenance along with food purposes for humans as well as animals [2]. Currently, in the food industry, preservation of food, quality maintenance, and safety are considered major growing concerns [3,4]. The bad quality of food can cause several infectious diseases, due to food contamination and spoilage which is highly increased due to the presence of bacteria and various fungi. The chief rewards of attaining drugs from a natural source are easy availability and relatively fewer side properties [5]. Due to the revolution of traditional culture, aboriginal knowledge of herbal plants is being decreased [1]. *Citrullus colocynthis* fruits produce the colocynthin alkaloid, and roots, as well as the fruit of this plant, are used to treat snake bites [6]. *Bryonia alba* and *Ecballium elaterium* have been used as homeopathic therapeutic agents [7]. The fruit and leaves of *Capsicum annuum* L. are being used for the reduction of pain due to dog bites and other skin problems [8]. The leaves of *Calotropis procera* are useful for treating jaundice [9]. *Solanum melongena* fruit is cardio-tonic while extracts of its leaves are being used against bronchitis, cholera, fever, and asthma; its roots are used for anti-inflammation along with anti-ulcer [8]. *Cannabis sativa* (Marijuana) has cannabinoids like cannabinol, cannabichromene, Δ9-tetrahydrocannabinol, and cannabigerol, having an antibacterial effect against *Staphylococcus aureus* stain [10]. Leaf extract of *Coccinia indica* gives membrane stabilizing, antimicrobial, antioxidant, and thrombolytic in vitro activities [11]. *Albizia myriophylla* was explored for α-glucosidase inhibition by in vitro model, and phytochemicals; 3, 4, 7, 3′-tetrahydroxyflavan, and 8-methoxy-7, 3′,4′-trihydroxyflavone and indenoic acid were accountable for an anti-diabetic effect [12]. *Cydonia oblonga* and *Allium porrum* can be used to treat patients of Type II diabetes mellitus [13]. *Bergenia ciliate* has two active compounds (-)-3-O-galloylepicatechin and (-)-3-O-galloylcatechin, responsible for inhibiting the digestive enzyme activities of α-glucosidase and α-amylase [14].

Nowadays interest in the uses of plant extract or their purified compounds has been increasing rapidly, especially in the scavenging of free radicals responsible for different diseases [15]. Reactive oxygen species (ROS) can effortlessly pledge the injury of the cell membrane components, especially phospholipids and lipoproteins, by spreading a free radicals reaction [16]. It is reported in the literature that the antioxidant potential of the plants is mainly due to their phenolic constituents [17]. In a longer period, plant families recognized as having higher antioxidant activity might be of worth in the design of advanced studies to unravel innovative dealing with illnesses related to free radicals. Several synthetic antioxidants, butylated hydroxyanisole (BHA), butylated hydroxytoluene (BHT), and tertiary butylhydroquinone (TBHQ) have been added to our junk food but, unfortunately, are reported to cause liver disorders [16]. Therefore, it is a need of this era to use less toxic and easy plant materials or their compounds purified from edible sources.

Diabetes mellitus disease is a metabolic disorder that causes high blood sugar because of insufficient insulin production. There are three types: type 1 [18], type 2 (non-insulin-dependent), and gestational diabetes [19]. Type 2 diabetes mellitus patients are 90%–95% of the total diabetes mellitus patients. The rate of diabetes estimated in 2000 was 2.8%, and it is estimated that it will be 4.4% in 2030. According to the WHO, diabetes will be the 7th leading cause of death in 2030 [20]. A lot of people have died because of contagious bacterial diseases both in developing and developed countries [18]. Microorganisms develop resistance against antibiotics with time. Synthetic drugs may be linked with severe effects on the host, involving allergic reactions, hypersensitivity, immune suppression, etc. Due to mankind’s nature of medicinal plant extracts, these compounds have obtained implausible achievements in helping find new antibiotics. For example, Quinine (cinchona) and Berberine (Berberis) are plant-derived antibiotics that are extremely auspicious against microbes like *E. coli* and *Staphylococcus auresus* [21]. Currently, medicinal plants are being used for the enrichment of functional food as these have phenolic compounds which are a tremendous source of antioxidants. These antioxidants are vital for our life sustainability as they scavenge the free radicals generated in our body due to environmental or other impacts. Food security is a global topic of this era, and we are dependent on these medicinal plants for our daily food. The plant extracts or their purified compounds are being used for increasing the shelf life of the various food products either in the form of antioxidants or antibacterial agents. Throughout the world, scientists are engaged in the identification and isolation of bioactive compounds from different species [22,23]. The present work was designed to explore the biological assessment of the plants in terms of antimicrobial, antioxidant and antidiabetics along with their secondary metabolites profiling. 

## 2. Materials and Methods

### 2.1. Chemicals and Instruments

The solvents used in extraction were of analytical grade and purchased from Panreac (Castellar del Vallès, Spain) while all other chemicals/reagents used for activities were of analytical grade and were purchased from Merck (Darmstadt, Germany). IR spectra were recorded on infrared spectrophotometer using ATR. 

### 2.2. Extraction of Plants Material

All plants (*Citrullus colocynthis*, *Solanum nigrum*, *Solanum surattense*, *Calotropis procera*, *Agave americana*, and *Anagallis arvensis*) were collected and identified by the taxonomist. After identification, plant materials were dried under shade and pulverized into powder. Powder from *C. colocynthis*, and *A. arvensis* were soaked in ethanol while *S. nigrum*, *S. surattense*, *A. americana* and *C. procera* were soaked in methanol for 15–20 days at room temperature, vigorously shaken at regular intervals, and filtered through vacuum filtration through a Büchner funnel. The solvent was evaporated from plant extracts by a rotary evaporator at reduced pressure at 40 °C, yielding a crude fraction. *A. arvensis*, and *A. americana* were extracted and got five fractions including aqueous according to scheme 1 (SS1). The aqueous solution of crude extract was partitioned with n-hexane, ethyl acetate, and 1-butanol. *C. colocynthis*, *S. nigrum*, *S. surattense*, and *C. procera* were extracted and portioned viz scheme 2 (SS2). Their crude extract was partitioned with chloroform [24,25]. Each fraction and extract was used in various studies (antimicrobial/antidiabetic) in the form of a solution.

### 2.3. DPPH Radical Scavenging Assay 

The antioxidant activity of crude extracts/fraction were measured in terms of the radical scavenging ability by the DPPH method [26]. Solution (1 mL) of all extracts/fraction at 100 mg/mL concentration was added to 2 mL methanolic solution of DPPH (2 mg/mL). The absorbance was noted at 517 nm after 30 min of incubation. The results were evaluated as the percentage scavenging of radicals using the reported formulae mentioned below. The results were compared with the standard (Gallic acid).
$$\mathrm{Inhibition}\;\left(\%\right)=\left(1-\frac{\mathrm{Absorbance}\;\mathrm{of}\;\mathrm{sample}}{\mathrm{Absorbance}\;\mathrm{of}\;\mathrm{control}}\right)\times 100$$

### 2.4. Antimicrobial Activity

To evaluate the antimicrobial activity of the plant extracts, antibacterial and antifungal activities were performed. For this purpose, 10 bacteria (*Klebsiella pneumonia*, *Bacillus subtilis*, *Staphylococcus aureus*, *Escherichia coli*, *Salmonella typhimurium*, *Halomonas salina*, *Neisseria gonorrhoeae*, *Shigella sonnei*, *Halomonas halophila*, and *Chromohalobacter israelensis*) and 2 fungi strains (*Aspergillus flavus* and *Aspergillus niger*) were selected on the availability of resources. All the plant extracts and fractions were screened in vitro for antimicrobial activities against bacterial and fungal strains. The medium was prepared, autoclaved, and poured into petri plates. Each petri plate was seeded with respective bacterial and fungal strains. A total of 20 µL of the sample was introduced on each disc with a micropipette. Streptomycin (5 mg/mL) was used as a positive reference for bacterial strains and Fluconazole (5 mg/mL) was used as a positive reference for fungal strains. After loading samples, cultures were allowed to grow for 24 h in an incubator at 37 °C for bacterial strains, and, for fungal strains, plates were incubated at 25 °C for 48 h. After reported time, the activity of the microbes was measured in millimeters and average activity and zone of inhibition were taken [25,27].

### 2.5. Antidiabetic Activity

α-Amylase inhibitory activity was carried out according to the reported method [28] with some modifications. A total of 200 µL of sample solution, 200 µL amylase (0.5 mg/mL), and 100 µL phosphate buffer (20 mM) of pH 6.9 were incubated at room temperature for 10 min. After 10 min, 100 µL of starch solution was added and again incubated at 25 °C for 20 min. After incubation, 100 µL of dinitrosalicylic acid (DNS) and 100 µL NaOH were added to the above solution mixture and test tubes were put into boiling water for about 20 min and then cooled at room temperature. A total of 1.0 mL of distilled water was added to the reaction mixture and the absorbance was measured at 540 nm. Each assay for all samples was carried out in triplicate. Percentage inhibition of all samples was calculated using the below-mentioned formula.
$$\mathrm{Inhibition}\;\left(\%\right)=\left(1-\frac{\mathrm{Absorbance}\;\mathrm{of}\;\mathrm{sample}}{\mathrm{Absorbance}\;\mathrm{of}\;\mathrm{control}}\right)\times 100$$

### 2.6. GC-MS and FTIR Analysis

GC-MS analysis was performed with a gas chromatogram (Shamzadu GC-2010) along with mass spectrometer 9 (Shamazdu AOC-201). Samples for GC-MS were prepared in n-hexane. The sample was introduced in non-polar capillary Colum SH-Rxi™-1MS by a split-less injection system. Helium gas of ultra-high purity was used with the 1 mL/min flow rate [29]. FTIR spectra of plant extracts were recorded at room temperature through Nicolet iS5 FTIR spectrometer having ATR/iD3 with argon horizontal cell (Thermo Scientific^®^, Waltham, MA, USA). 

### 2.7. Statistical Analysis

All experiments were performed in triplicate formation and standard deviation (±SD) was calculated against each assay using the MS Excel program. 

## 3. Results and Discussion

The various parts of different plant species were collected and extracted in different solvents. The crude extracts were further partitioned with different solvents based on polarity with the solvent extraction technique (Table 1). The crude extracts and their respective fractions were subjected to phytochemical analysis using reported protocols which suggested that these plant materials are a rich source of secondary metabolites.

### 3.1. FT-IR Analysis of Plant Extracts

Identification of functional groups present in the studies of plant extracts is necessarily for the structure elucidation of the bioactive compounds. To find the functionality of the compounds in the crude extract and fraction, the FTIR technique was used, and functional moieties were identified based on different bands respective to their functional groups. The absorption band at 3550–3450 cm^−1^ showed the characteristic band of aromatic OH group in chelate while OH of primary and tertiary alcohol absorbed at 1350–1260 cm^−1^ and 1410 cm^−1,^ respectively. Fundamental stretching frequencies of the C-N bond appeared in the range of 1200–1000 and 1350 cm^−1^ to 1280 cm^−1^, and N-H absorbed at 3400–3380 and 3510–3460 cm^−1^ for aliphatic and aromatic amines, respectively. The bands at 1150–1060 cm^−1^, 1270–1230 cm^−1^, and 1250–1150 cm^−1^ are representing the presence of C-O (stretch) of dialkyl, alkyl, aryl, and diaryl ethers, respectively. Oximes can be confirmed by comparing band frequency at 1690–1590 cm^−1^ representing the presence of C=N and also broad OH stretch at 3300–3150 cm^−1^. The ʋ(-CH_3_) frequencies range from 1470–1430 cm^−1^ representing antisymmetric vibrations and 1380–1370 cm^−1^ representing symmetric vibration of ʋ(-CH_3_). The CH_2_ of R_2_C-CH_2_ vibrates at the frequency range from 3090 to 3075 cm^−1^, for in-plane CH_2_ vibrates at 1420–1410 cm^−1^, and for out-plane, CH_2_ vibration occurs at 895–885 cm^−1^ (Appendix A). The IR bands at 3463, 3439, 3442, 3424, and 3423 cm^−1^ are due to OH/NH, while bands at 1647, 1649, and 1652 cm^−1^ are due to carbonyl functionality in the compounds. The bands near 2900 cm^−1^ are indicated the presence of an aromatic C-H bond in the molecules. Similarly, bands ranged 1052–1062 cm^−1^ because of C-O-C (ether) linkage. Based on the FT-IR analysis results, it was depicted that these compounds have various functional moieties such as C=O, -OH, C-O, and C=C. From the results, it is analyzed that Oxime, ester, amide, carboxylic acid, and ether functional groups are present in most plant extracts (Table 2, Table 3 and Table 4).

### 3.2. GC-MS Analysis

GC-MS (gas chromatography- mass spectrometry) technique is an excellent technique used to identify unknown compounds present in plant extracts, drug metabolites, lipids, and environmental contaminants. Plant extract was completely dissolved in n-hexane, filtered through micro filter paper, and subjected to GC-MS analysis. Electronic voltage of −70 eV was given to the compound to fragmentize it [30]. The chromatogram of the n-Hexane fraction of *A. americana*, *A. arvensis,* and the chloroform fraction of *C. colocynthis* is presented in Figure 1, Figure 2 and Figure 3. GC-MS analysis of *A. americana* showed the presence of 11,14,17-eicosatetraene (RT = 16.260), 9,12,15-Octadecatriene-1-ol (RT = 11.974), Dodecanoic acid, 2-butoxyethyl ester (RT = 14.614), and 11,14,17-Eicosatrienoic acid (RT = 11.558). Hexadeanoic acid (RT = 2.063) is present in the n-hexane fraction of *A. americana*. GC-MS analysis of *A. arvensis,* the n-hexane fraction revealed that 1,2-Benzenedicarboxylic acid and Octane are present in this fraction at retention times of 15.646 and 2.057 min, respectively. GC-MS analysis of the chloroform fraction of *C. colocynthis* showed four compounds; 1, 2-Benzenedicarboxylic acid, 9, 12-Octadecadienoic acid, 11-Octadecadienoic acid, phytol, and hexadeanoic acid at 15.633, 11.487, 11.529 and 9.733 min, respectively.

### 3.3. Antioxidant Activity

The medicinal plants are a reliable source of antioxidant moieties being used for a long time as a crude extract as well as their purified compounds. The community of developing countries and people living in rural areas are still dependent on such therapeutic agents for their health issues. In the present study, we have extracted different plants in ethanol/methanol which were further partitioned with various solvents. The extracts/fractions were screened for their antioxidant potential using DPPH as a free radical. It was depicted that most of the plant fusion has antioxidant potential as mentioned in Table 5. The maximum activity was exhibited by AAEA while the minimum potential was shown by AARH (15.3 ± 0.1%), which may be due to the presence of low molecular weight compounds like a hydrocarbon. The highest activity of the AAEA may be due to polar components in the fraction (78.1 ± 0.3%). It was found that aqueous fractions in most of the cases remained less effective against a tested activity as depicted in Figure 4. The polar solvent has polar compounds such as phenolic acid or flavonoids which have great potential to quench the free radicals. Our findings about the antioxidant potential of understudy compounds are lined with already reported literature that medicinal plants have secondary metabolites that are responsible for antioxidant activities [31,32,33,34].

### 3.4. Antimicrobial Activity

#### 3.4.1. Antibacterial Activity

The crude extracts along their fractions (solvent extraction) were subjected to antibacterial assessments. The extracts of *C. colocynthis*, *S. nigrum*, *S. surattense*, *C. procera*, *A. americana*, *A. arvensis*, and *B. hispida* were evaluated against 10 bacterial strains as mentioned in the experimental section. The ethanol fraction of *C. colocynthis*, the chloroform fraction of fruit of *S. surattense*, the methanol fraction of *A. americana*, and the n-hexane fraction of *A. americana* inhibited most of the bacteria. The standard (STD) used against these bacteria was Streptomycin. Ethanol, methanol, n-hexane, and chloroform fractions of plant extracts were mostly active against these bacteria. The variations in the antibacterial potential of plant extracts are due to the different chemical compositions of different extracts. The highest value exhibited against *E. coli* by AAH was 21 mm; against *K. pneumonia* by SSFC it was 30 mm; against *B. subtilis* by CCE it was 15 mm; against *S. aureus* by CCE it was 32.5 mm; against *S. typhimurium* by AAH it was 20 mm; against *C. israelensis* by CCE it was 25 mm; against *H. salina* by AARH it was 24 mm; against *N. gonorrhoeae* by AAA it was 21 mm; against *S. sonnei* by AAM it was greater against *H. halophile* (Table 6). SSFC represented higher activity than exhibited by the standard’s activity against *K. pneumonia* strain; CCE exhibited activity of 32 mm against *Staphylococcus aureus*, which is very close to the activity value (35 mm) of the standard. Similarly, the order of the activity was AARH > AAM > AAA against *N. gonorrhoeae*, *S. sonnei*, and *H. salina.* Strains were similar or close to the antibacterial activity exhibited by the standard against these bacterial strains (Table 6). Based on FTIR and GCMS results, it was concluded that this fraction has hydroxyl-containing compounds which may be responsible for antimicrobial activity. Natalia et al. (2021) reported that phenolic composites are diverse groups of natural products present in medicinal plants, which may be widely employed against various disease-causing microbes [35]. In literature, it was reported that secondary metabolites such as saponins, alkaloids, and phenolics have proven potential medicinal impacts for the cure of many diseases such as heart-related illnesses, cancers, bacterial infections, tumors, and diabetes [24,36,37].

#### 3.4.2. Antifungal Activity

Antifungal activity of all plant extracts was tested against *Aspergillus flavus* and *Aspergillus niger* estimated via disc diffusion method. Fluconazole was used as the standard (STD) against these fungal strains. The zone of inhibition was measured after 48 h and was incubated at 25 °C. The chloroform fraction of *S. nigrum,* the ethanolic fraction of *A. americana, the* ethanolic fraction of leaves and stem of *S. surattense*, and the chloroform fraction of leaves and stem of *Solanum surattense* showed significant activity against *A. flavus*. The butanolic fraction of *A. arvensis*, the hexane fraction of *A. arvensis*, the methanolic fraction of *A. americana*, the n-hexane fraction of *A. americana*, and the aqueous fraction of *C. colocynthis* exhibited good activity against *A. niger* fungi. The highest value of inhibition against *A. flavus* was exhibited by SNC and was 21 mm. Against *A. niger* the highest value was exhibited by AARB and was measured at 24 mm. Values of antifungal activity of SNC, AARB, and AARH were higher than that of the standard’s value of inhibition, showing that these fractions have a higher potential to inhibit fungal growth. Results indicated chloroform fractions are mostly active against these fungal strains (Figure 5). Infections due to fungal species are increasing day by day and are one of the main reasons for deaths [38]. The fungi which are toxic to human beings comprise *Fusarium, Aspergillus,* and *Candida* species; these species secrete hazardous materials such as mycotoxins. To minimize or protect from fungal infections, plants secret many types of secondary metabolites, which perform vital roles as protection against several infections. The use of plant-based therapeutic agents for different ailments is preferred by the local community due to its fewer side effects and excellent activity [39].

#### 3.4.3. Antidiabetic Activity

The antidiabetic activity determined via the inhibition of α-amylase enzyme studies showed that the chloroform fraction of *C. colocynthis*, *S. nigrum* and *S. surattense* (leaves), the *S. surattense* aqueous fraction of leaves and stem, the *C. procera* methanol fraction, the *C. procera* chloroform fraction, the *A. arvensis* ethanol fraction, the *A. arvensis* hexane fraction, *the A. arvensis* ethyl acetate fraction, the *A. americana* ethyl acetate fraction, the *A. americana* butanol fraction, and the *A. americana* aqueous fraction showed good inhibition against α-amylase. It is observed that chloroform fractions, ethanol fractions, and hexane fractions were mostly active against α-amylase (Figure 6). These fractions have a rich source of hydroxyl and nitrogen containing compounds in the form of alcohol/phenol and amide or amines form which have the tendency to inhibit the targeted enzyme due to the strong binding on the active site of the enzyme.

The highest value of α-amylase inhibition was depicted by the chloroform fraction of *S. surattense* (SSFC) at 89.56%, while the chloroform fraction of *C. procera* has shown weak response. The highest inhibition among ethanolic extract was shown by AARE at 95.65% and the fraction of *S. surattense* exhibited low inhibition against diabetic enzymes. The aqueous fraction *A. arvensis* (AAA) represented the highest anti-diabetic activity (94.53%) while *C. colocynthis* had about no activity as presented in Figure 6. The antidiabetic results of the selected medicinal plants are tabulated in Table 7. Based on the results, it is depicted that understudied plant extracts/fractions have excellent potential for antidiabetic activity. The variation in the activity may be due to differences in qualitative and quantitative phytochemicals.

Natural products chiefly derived from a plant source are also major excavations for lead compounds and play an imperious part in new drug discovery [40,41]. The rural community is mainly dependent on the natural product due to cheaper prices, fewer side effects, and easy availability. The plant-based materials are the key player in all available therapies, especially in rural areas. They keep dominant pharmacological activities to ward off many diseases [42]. Many plants are a rich source of different therapeutic agents necessary for various diseases and, currently, several drugs on the market are either natural products or their derivatives. The people of developing areas have placed extraordinary expectations on these plant behaviors, and the consumption of economic medicinal plants instead of drugs to cure diabetes is common. Medicinal plants have numerous phytoconstituents (e.g., saponins, flavonoids, terpenoids, carotenoids, glycosides, alkaloids) with antidiabetic activity [43]. The intricate plant medium is an exporter of several phytoconstituents, which defines the exact communication of these compounds. Chan et al. (2012) reported that compounds used against diabetics are mainly found in the aerial parts of the plants [44] and our findings are in agreement with this study that leaves, fruit and whole plants are effective against diabetics.

## 4. Conclusions

Six plants species were targeted for their biological evaluation in terms of antibacterial, antifungal, antioxidant and antidiabetic properties. FT-IR and GC-MS analyses were carried out to estimate the secondary metabolites of each extract/fraction. The FT-IR results indicated the presence of acids, amide, ester, and ether functional groups, while GC-MS analysis revealed the presence of carboxylic acid, alkenes, and ester moieties in the understudied plants. The results showed that the ethanoic fraction of *C. colocynthis* inhibited most of the bacterial strains, while SSFC, AAM, and AAH fraction inhibited 80% of bacteria; against fungal strains, AARB and SNC remained most active. AARB and AAREA exhibited the highest anti-diabetic activity, while 70 percent plant extract/fractions depicted good antioxidant potential. It is concluded from the results that the plants used for this study are a rich source of bioactive agents and can be used against tested activities after their complete secondary metabolites test along with complete in vitro and in vivo assessments. Further, docking studies, structure elucidation, Quantitative structure-activity relationship (QSAR), and structure-activity relationship (SAR) of the secondary metabolites present in these plants should be studied.

## Figures and Tables

**Figure 1 molecules-27-05935-f001:**
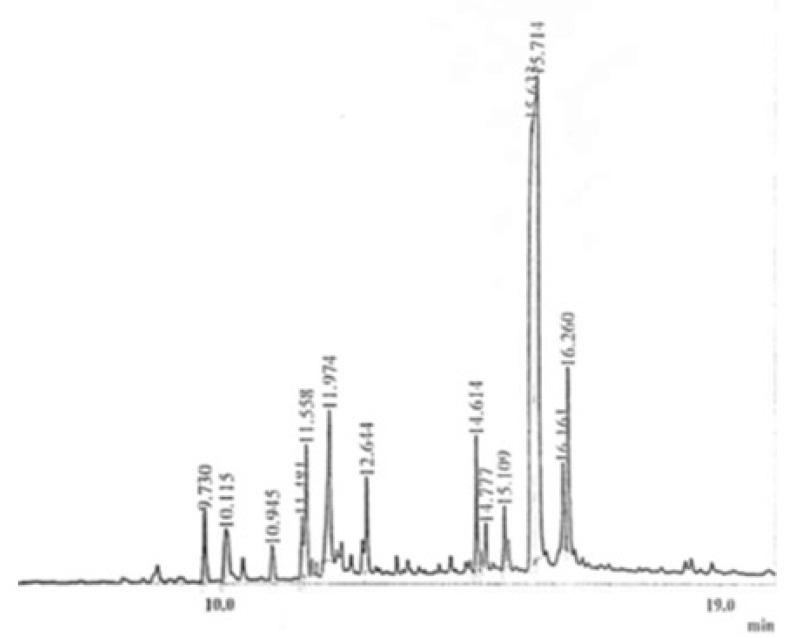
GC-MS chromatogram of n-Hexane fraction of *A. americana.*

**Figure 2 molecules-27-05935-f002:**
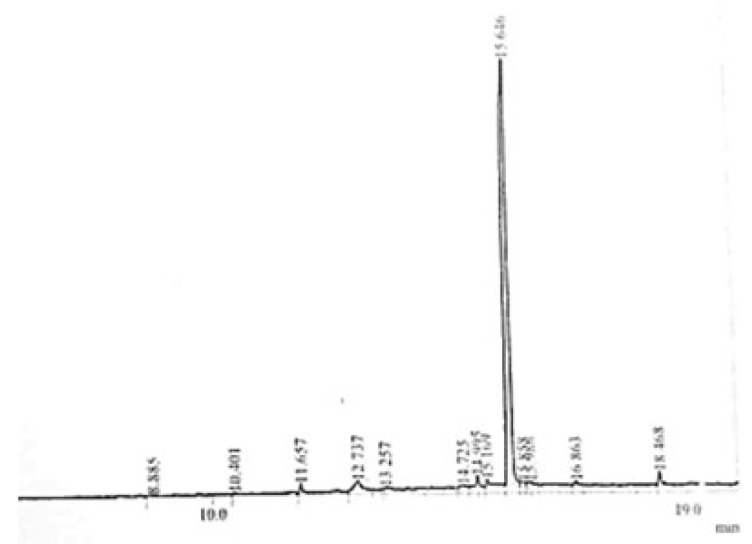
GC-MS chromatogram of n-Hexane fraction of *A. arvensis.*

**Figure 3 molecules-27-05935-f003:**
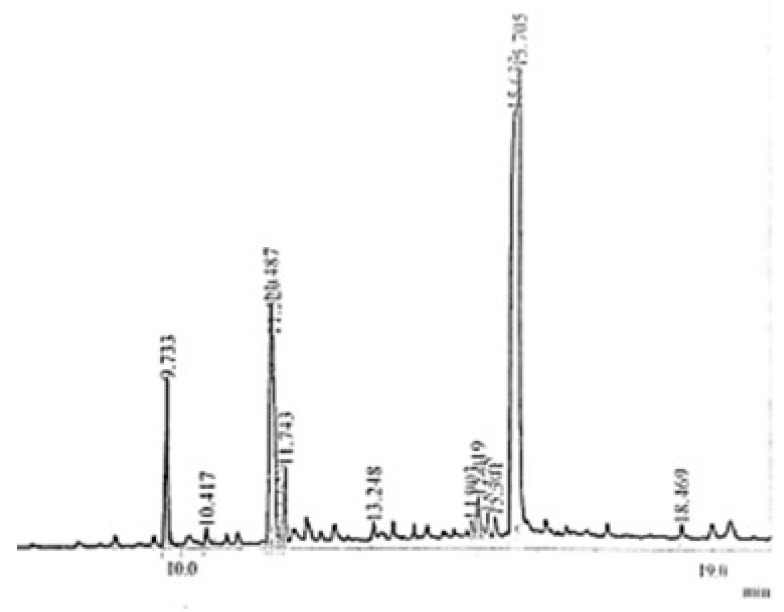
GC-MS chromatogram of chloroform fraction of *C. colocynthis.*

**Figure 4 molecules-27-05935-f004:**
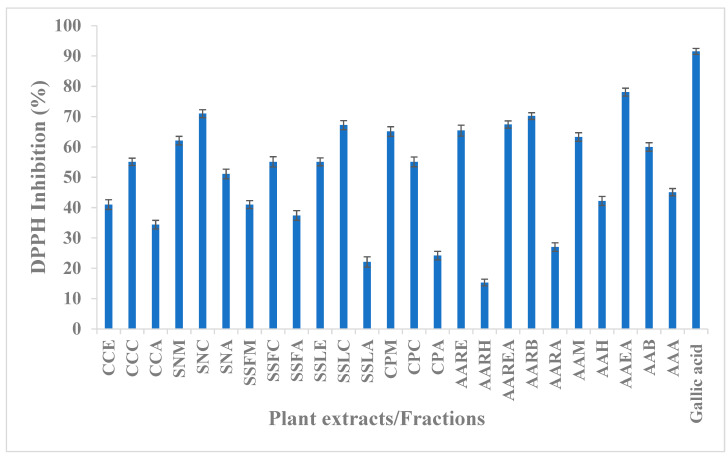
Graph of antioxidant activity of plant extracts.

**Figure 5 molecules-27-05935-f005:**
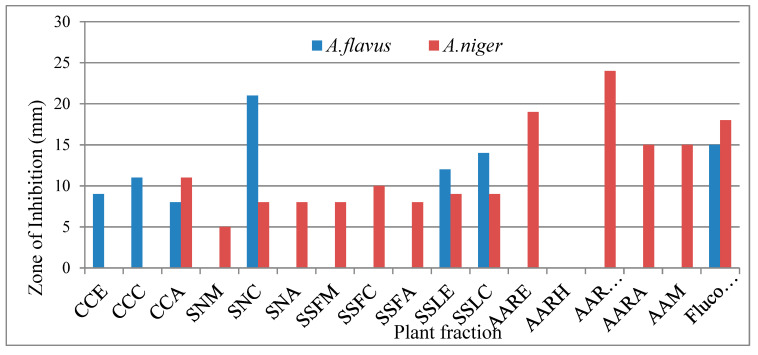
Graph of antifungal activity of plant extracts.

**Figure 6 molecules-27-05935-f006:**
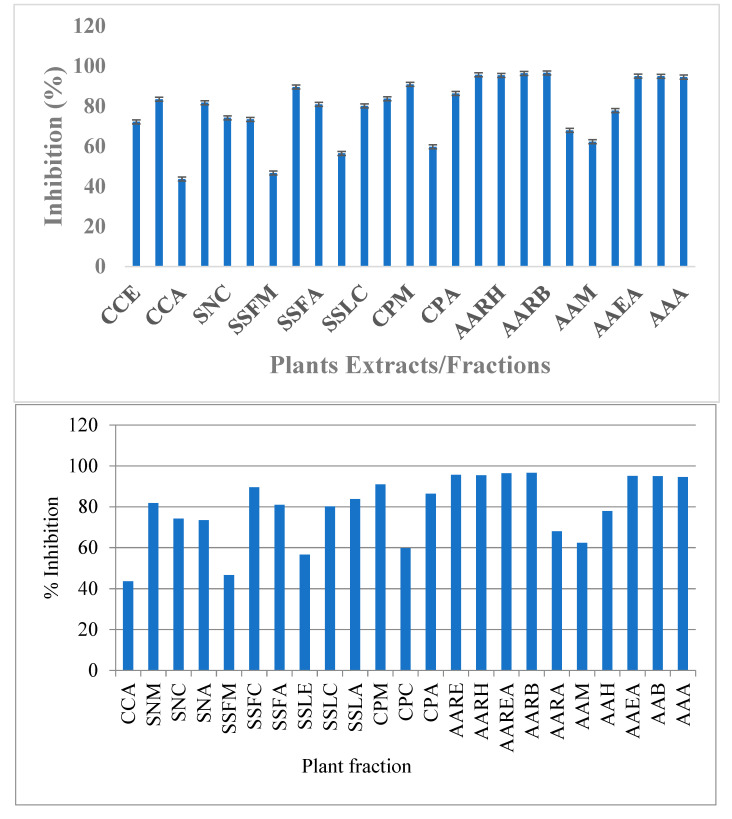
Graph of antidiabetic activity of plant extracts.

**Table 1 molecules-27-05935-t001:** Medicinal plants selected for the current study.

Species Name	Family	Part Used	Solvent	Fraction	Code
*Citrullus colocynthis*	Cucurbitaceae	Fruit	Ethanol	Crude	CCE
				Chloroform	CCC
				Aqueous	CCA
*Solanum nigrum*	Solanaceae	Whole plant	Methanol	Crude	SNM
				Chloroform	SNC
				Aqueous	SNA
*Solanum surattense*	Solanaceae	Leaves	Methanol	Crude	SSFM
				Chloroform	SSFC
				Aqueous	SSFA
		Fruit	Ethanol	Crude	SSLE
				Chloroform	SSLC
				Aqueous	SSLA
*Calotropis procera*	Asclepiadaceae	Leaves	Methanol	Crude	CPM
				Chloroform	CPC
				Aqueous	CPA
*Agave americana*	Asparagaceae	Leaves	Ethanol	Crude	AARE
				n-Hexane	AARH
				Ethyl acetate	AAREA
				Butanol	AARB
				Aqueous	AARA
*Anagallis arvensis*	Primulaceae	Whole plant	Methanol	Crude	AAM
				n-Hexane	AAH
				Ethyl acetate	AAEA
				Butanol	AAB
				Aqueous	AAA

**Table 2 molecules-27-05935-t002:** Fundamental IR vibrations of phytochemicals present in *S. surrattense* fractions.

Plant Fraction							
SSLC		SSFC		SSLA		SSFA	
Wave Number (cm^−1^)	Functional Group	Wave Number (cm^−1^)	Functional Group	Wave Number (cm^−1^)	Functional Group	Wave Number (cm^−1^)	Functional Group
3463	OH/NH	3442	OH/NH	3424	NH amide	3423	N-H amide
1647	C=O (amide)	1647	C=O (amide)	1649	C=O (amide)	1649	C=O (amide)
1436	C-H	1415	C-H	1437	C-H	1438	C-H
1386	-OH	1387	-OH	1386	-OH	1386	-OH
1252	Aryl ether	1253	Aryl ether	1254	Aryl ether	1254	Aryl ether
1100	C-O-C acyclic	1101	C-O-C acyclic	1062	C-O-C acyclic	1062	C-O-C acyclic
848	C-N aromatic	1060	C-N amine aliphatic	1094	C-N amine aliphatic		

**Table 3 molecules-27-05935-t003:** Fundamental IR vibrations of phytochemicals present in the aqueous fraction of *C. procera*, *C. colocynthis,* and *A. arvensis.*

CPA		CCA		AARH	
Wave Number (cm^−1^)	Functional Group	Wave Number (cm^−1^)	Functional Group	Wave Number (cm^−1^)	Functional Group
3439	N-H (amide)	3225	N-H (amide)	3412	N-H (amide)
2930	C-H (aromatic)	2930	C-H (aromatic)	2992	-OH (COOH)
1649	C=O (amide)	1652	C=O (amide)	1714	C=O (ester)
1437	-CH_2_-	1436	-CH_2_-	1655	C=O (amide)
1386	-OH (3*)	1387	-OH (3*)	1438	-CH_2_-
1254	Aryl ether	1253	Aryl ether	1254	Aryl ether
1094	C-O (aliphatic)	1094	C-O (aliphatic)	1095	C-O (aliphatic)
1056	C-O-C dialkyl ether	1163	C-O- diaryl ether	1062	C-N (amine)

**Table 4 molecules-27-05935-t004:** Fundamental IR vibrations of phytochemicals present in fractions of *A. americana.*

AAEA		AAM		AAH	
Wave Number (cm^−1^)	Functional Group	Wave Number (cm^−1^)	Functional Group	Wave Number (cm^−1^)	Functional Group
3423	NH amine	3423	OH (COOH)	3425	NH amine
2928	Ac-H carbonyl	2930	Ac-H carbonyl	2922	Ac-H carbonyl
1649	C=O amide	1648	C=O amide	1652	C=O amide
1438	-CH_2_-	1438	-CH_2_-	1438	-CH_2_-
1411	RCH=CH_2_	1411	RCH=CH_2_	1411	RCH=CH_2_
1387	-OH (alcohol)	1387	-OH (alcohol)	1386	OH (alcohol)
1254	Aryl ether	1254	Aryl ether	1254	Aryl ether
1095	C-O (aliphatic)	1096	C-O (aliphatic)	1095	C-O (aliphatic)
1062	C-O-C (acyclic)	1061	C-O-C (acyclic)	1061	C-O-C (acyclic)

**Table 5 molecules-27-05935-t005:** Antioxidant activity of plant extracts.

Code	% Inhibition	Code	% Inhibition
CCE	41.0 ± 1.6	CPC	55.1 ± 1.6
CCC	55.1 ± 1.2	CPA	24.2 ± 1.4
CCA	34.4 ± 1.4	AARE	65.4 ± 1.8
SNM	62.1 ± 1.7	AARH	15.3 ± 1.1
SNC	71.0 ± 1.3	AAREA	67.4 ± 1.2
SNA	51.1 ± 1.6	AARB	70.2 ± 1.1
SSFM	41.0 ± 1.3	AARA	27.0 ± 1.4
SSFC	55.1 ± 1.7	AAM	63.3 ± 1.4
SSFA	37.4 ± 1.6	AAH	42.2 ± 1.5
SSLE	55.1 ± 1.3	AAEA	78.1 ± 1.3
SSLC	67.2 ± 1.5	AAB	60.0 ± 1.4
SSLA	22.1 ± 1.7	AAA	45.1 ± 1.2
CPM	65.1 ± 1.6	Gallic acid (STD)	91.5 ± 1.0

**Table 6 molecules-27-05935-t006:** Antibacterial activities shown by different fractions of plant extracts.

Strains/Codes	Zone of Inhibition (mm)									
	EC	KP	BS	SA	ST	CI	HS	NG	SS	HH
CCE	**20 ± 2**	**15±**	**15 ± 1**	**32 ± 2**	**13 ± 1**	**25 ± 2**	**0 ± 0**	**13 ± 1**	0 ± 0	0 ± 0
CCC	0 ± 0	0 ± 0	0 ± 0	0 ± 0	0 ± 0	0 ± 0	0 ± 0	0 ± 0	0 ± 0	0 ± 0
CCA	0 ± 0	0 ± 0	0 ± 0	0 ± 0	0 ± 0	0 ± 0	0 ± 0	0 ± 0	0 ± 0	0 ± 0
SNM	11 ± 1	10 ± 1	**8 ± 1**	0 ± 0	**9 ± 1**	**17 ± 1**	0 ± 0	0 ± 0	0 ± 0	0 ± 0
SNC	0 ± 0	0 ± 0	0 ± 0	0 ± 0	0 ± 0	0 ± 0	0 ± 0	10 ± 1	0 ± 0	0 ± 0
SNA	0 ± 0	0 ± 0	0 ± 0	0 ± 0	0 ± 0	0 ± 0	0 ± 0	**12 ± 2**	0 ± 0	0 ± 0
SSFM	0 ± 0	0 ± 0	0 ± 0	0 ± 0	0 ± 0	**11 ± 1**	0 ± 0	13 ± 2	0 ± 0	0 ± 0
SSFC	**8 ± 1**	**30 ± 1**	**10 ± 1**	0 ± 0	**19 ± 1**	11 ± 2	12 ± 2	**15 ± 1**	0 ± 0	**11 ± 2**
SSFA	0 ± 0	0 ± 0	0 ± 0	0 ± 0	0 ± 0	0 ± 0	**13 ± 2**	13 ± 2	0 ± 0	0 ± 0
SSLE	0 ± 0	0 ± 0	0 ± 0	0 ± 0	0 ± 0	**13 ± 1**	0 ± 0	0 ± 0	0 ± 0	0 ± 0
SSLC	0 ± 0	**12 ± 1**	0 ± 0	0 ± 0	10 ± 1	0 ± 0	12 ± 1	12 ± 1	0 ± 0	0 ± 0
SSLA	0 ± 0	0 ± 0	0 ± 0	0 ± 0	0 ± 0	0 ± 0	13 ± 1	11 ± 1	0 ± 0	0 ± 0
CPM	0 ± 0	0 ± 0	0 ± 0	0 ± 0	0 ± 0	**15 ± 1**	0 ± 0	11 ± 1	0 ± 0	0 ± 0
CPC	0 ± 0	0 ± 0	**12 ± 1**	0 ± 0	0 ± 0	0 ± 0	0 ± 0	11 ± 1	0 ± 0	0 ± 0
CPA	0 ± 0	0 ± 0	0 ± 0	0 ± 0	0 ± 0	0 ± 0	**14 ± 1**	**13 ± 1**	**14 ± 1**	0 ± 0
AARE	0 ± 0	0 ± 0	0 ± 0	0 ± 0	0 ± 0	0 ± 0	12 ± 1	11 ± 2	**5 ± 1**	11 ± 1
AARH	0 ± 0	0 ± 0	0 ± 0	0 ± 0	0 ± 0	**19 ± 1**	**24 ± 2**	12 ± 1	0 ± 0	0 ± 0
AAREA	0 ± 0	0 ± 0	0 ± 0	0 ± 0	0 ± 0	11 ± 1	10 ± 1	13 ± 1	0 ± 0	**12 ± 1**
AARB	0 ± 0	0 ± 0	0 ± 0	0 ± 0	0 ± 0	0 ± 0	17 ± 1	**15 ± 2**	**21 ± 2**	0 ± 0
AARA	**15 ± 2**	0 ± 0	0 ± 0	0 ± 0	0 ± 0	0 ± 0	11 ± 1	10 ± 1	0 ± 0	0 ± 0
AAM	20 ± 1	0 ± 0	**13 ± 1**	0 ± 0	13 ± 1	10 ± 1	20 ± 1	10 ± 1	**22 ± 2**	**14 ± 2**
AAH	**21 ± 1**	**10 ± 1**	0 ± 0	**13 ± 1**	**20 ± 2**	0 ± 0	**21 ± 1**	**22 ± 2**	0 ± 0	11 ± 1
AAEA	10 ± 1	0 ± 0	0 ± 0	0 ± 0	0 ± 0	0 ± 0	16 ± 1	20 ± 2	0 ± 0	10 ± 2
AAB	0 ± 0	0 ± 0	0 ± 0	0 ± 0	0 ± 0	9 ± 1	**22 ± 2**	0 ± 0	0 ± 0	0 ± 0
AAA	0 ± 0	0 ± 0	0 ± 0	0 ± 0	0 ± 0	**10 ± 1**	12 ± 1	21 ± 2	0 ± 0	0 ± 0
STD	30 ± 1	26 ± 2	22 ± 1	35 ± 2	28 ± 1	32 ± 1	27 ± 1	23 ± 2	22 ± 2	32 ± 2

EC = Escherichia coli; KP = Klebsiella pneumonia; BS = Bacillus subtilis; SA = Staphylococcus aureus; ST = Salmonella typhimurium; CI = Chromohalobacter israelensis; HS = Halomonas salina; NG = Neisseria gonorrhoeae; SS = Sheigella sonnei; and HH = Halomonas halofila.

**Table 7 molecules-27-05935-t007:** Anti-diabetic activity of plant extracts.

Code	% Inhibition	Code	% Inhibition
CCE	72.23 ± 1.4	CPC	59.76 ± 1.9
CCC	83.45 ± 1.5	CPA	86.35 ± 1.7
CCA	43.65 ± 1.3	AARE	95.65 ± 2.2
SNM	81.76 ± 1.4	AARH	95.40 ± 2.4
SNC	74.23 ± 1.9	AAREA	96.40 ± 1.3
SNA	73.45 ± 1.4	AARB	96.63 ± 2.1
SSFM	46.65 ± 1.1	AARA	67.95 ± 1.0
SSFC	89.56 ± 2.1	AAM	62.29 ± 1.1
SSFA	80.90 ± 1.7	AAH	77.87 ± 1.4
SSLE	56.50 ± 1.1	AAEA	95.05 ± 2.0
SSLC	80.15 ± 1.3	AAB	94.97 ± 2.7
SSLA	83.76 ± 1.7	AAA	94.53 ± 1.6
CPM	90.90 ± 1.2		

## Data Availability

Not applicable.

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
