# Peer review of "Extraction of Bioactive Compounds for Antioxidant, Antimicrobial, and Antidiabetic Applications"

_molecules, 2022, doi:10.3390/molecules27185935_

Round 1

Reviewer 1 Report

Extraction of Bioactive Compounds from medicinal plants for antioxidant, antimicrobial, and antidiabetic applications

The authors selected Citrulus colocynthis, Solanum nigrum, Solanum surattense, Calotropis procera, Agave americana, and Anagallis aroensis for studies on antioxidants, antibacterials, antifungals, and antidiabetic agents. The data is detailed and the experimental workload is large, and the structure-activity relationship between the structure and activity of phytochemicals is described more completely. However, there are the following problems that need to be modified:

1. Materials and methods need to be supplemented with more detailed chemical reagent purity and manufacturer sources.

2. The author should improve the quality of figure 1-13.

3. Figure 15 and 18 - Authors should add error bars to the chart.

4. DPPH antioxidant activity experiments should be performed with VC added as a positive control.

5. Lines 341-350: in the antidiabetic results discussion section it is hoped that the authors, in combination with characterization such as infrared groups or other demonstrated pathways, give different active ingredient inhibition α-amylase enzyme Mechanism to demonstrate the size order of different active ingredients in antidiabetic ability.

6. Line 330 requires format modification.

7. The format of the full text and the layout of the article need to be carefully revised.

Author Response

We are thankful to the referee for providing valuable comments which will be helpful in polishing the manuscript. We revised our manuscript according to their suggestions and attached the response to each comment.

Reviewer 2 Report

Review comments to the author

Title: ''Extraction of Bioactive Compounds from medicinal plants for antioxidant, antimicrobial, and antidiabetic applications''.

Manuscript ID: molecules-1889455.

1. Introduction:

1- Page 1, Lines 33-35: The section ''The bad quality of food can cause several infectious  diseases, due the food contamination and spoilage which is highly increased due to presence of  bacteria and various fungi'' should be replaced by another one since it was already mentioned in the abstract.

2- Page 2, Line 55: The word ''Now a day'' should be written as ''Nowadays''. 

3- Page 2, Lines 80-90: This section should be reinforced by relevant citations.

4- Page 2, Line 80: The word ''Now a day'' should be written as ''Nowadays''.

5- Page 2, Line 91: The word ''in-vitro'' should be written in italic font.

6- The introduction is relatively large and should be shortened.  

2. Materials and methods:

1- Add a new section under the title ''Plant material'', it should contains the collected parts from each plant under investigation, the place and time of collection, the name of the plant identifier, and the voucher number for each sample.

2.1. Extraction of plants material

1- Page 3, Line 108: The palnt names ''A. arvensis'' & ''A. Americana'' should be written in italic fonts.

Table 1:

1- Page 3, Line 114: The title of the first column ''Specie name'' should be written as ''Species name''.

Schem 2:

1- Page 4, Lines 119-120: The palnt names ''C. colocynthis, S. nigrum, S. surattense, & C. 119 procera'' should be written in italic fonts.

2- Page 4, Line 120: Delete the repeated dot (.).

Statistical analysis:

1- Insert a new section under the name ''Statistical analysis''.

Resutls and Discussion:

1- Page 10, Line 211: The title of figure 5 was hidden.

2- I suggest moving the figures (1-10) to the supplementary file.

3- The resolution and the quality of all GC-MS chromatograms should be improved.

4- Title of figure 15 should be written as ''Graph of antioxidant  activity of plant extracts.''

3.4. Antimicrobial activity

3.4.1. Antibacterial activity

1- Page 17, Lines 306-308: All names of the microbial species under table 6 should be written in italic fonts.

3.4.3. Anti-diabetic activity

1- The anti-diabetic activity should be compard with standard antidiabetic agent.

Abbreviations:

- List of abbreviations should be inserted by the end of the manuscript before references.

References:

1- All scientific names of plants and species should be written in italic fonts.

2- Journal names should be written in a uniform manner, either in complete from or abbreviated.

Author Response

(The authors gave the same response as above.)

Reviewer 3 Report

The research is very extensive and interesting, it is recommended to improve the presentation of the results to make them more understandable and easy to visualize. It is recommended to continue with the research to elucidate the secondary metabolites responsible for the biological activity, as well as to evaluate the toxicity of the bioactive fractions for their safe use.

Write the conclusions in a general way regarding the objective of the investigation and mention the most outstanding results and the greatest biological activity of each plant.

Author Response

(The authors gave the same response as above.)

Reviewer 4 Report

This paper does not provide clarity on the related field, very vaguely written paper that require complete re-writing from abstract to discussion. It must not be accepted for publication in this form. 

Please see all the comments provided in the manuscript and correct them accordingly. 

Special work on English, Quality of figures and arrangement of reference is also required. 

Author Response

(The authors gave the same response as above.)

Round 2

Reviewer 4 Report

English language of the paper is not improved substantially, therefore, I strongly recommend professional English editing. Overall quality of manuscript is improved.